# Study on the Printability of Starch-Based Films Using Ink-Jet Printing

**DOI:** 10.3390/ma17020455

**Published:** 2024-01-18

**Authors:** Zuzanna Żołek-Tryznowska, Katarzyna Piłczyńska, Tomasz Murawski, Arkadiusz Jeznach, Krzysztof Niczyporuk

**Affiliations:** 1Faculty of Mechanical and Industrial Engineering, Warsaw University of Technology, Narbutta 85, 02-524 Warsaw, Poland; katarzyna.pilczynska@pw.edu.pl (K.P.); tomasz.murawski.dokt@pw.edu.pl (T.M.); arkadiusz.jeznach@pw.edu.pl (A.J.); 2Association of Polish Engineers and Mechanical Technicians, Czackiego 3/5, 00-043 Warsaw, Poland; krzysztof.niczyporuk@simp.pl

**Keywords:** starch films, ink-jet printing, print quality, QR code, packaging

## Abstract

Starch-based films are a valuable alternative to plastic materials that are based on fossil and petrochemical raw resources. In this study, corn and potato starch films with 50% glycerol as a plasticizer were developed, and the properties of films were confirmed by mechanical properties, surface free energy, surface roughness, and, finally, color and gloss analyses. Next, the films were overprinted using ink-jet printing with quick response (QR) codes, text, and pictograms. Finally, the print quality of the obtained prints was determined by optical density, color parameters, and the visual evaluation of prints. In general, corn films exhibit lower values of mechanical parameters (tensile strength, elongation at break, and Young Modulus) and water transition rate (11.1 mg·cm^−2^·h^−1^) than potato starch film (12.2 mg·cm^−2^·h^−1^), and water solubility is 18.7 ± 1.4 and 20.3 ± 1.2% for corn and potato film, respectively. The results obtained for print quality on starch-based films were very promising. The overprinted QR codes were quickly readable by a smartphone. The sharpness and the quality of the lettering are worse on potato film. At the same time, higher optical densities were measured on potato starch films. The results of this study show the strong potential of using starch films as a modern printing substrate.

## 1. Introduction

There is a growing demand for ecologically sustainable alternatives to traditional plastic packaging. The researchers seek novel, functional, environmentally friendly, or biobased materials. The research on biodegradable and renewable materials follows the UE directive [1]. Starch is abundant, biodegradable, biocompatible, cheap, and renewable. Also, it should be highlighted that it can be harvested without destroying the plant [2,3]. Due to their excellent filmogenic properties, low cost, and biodegradation, thermoplastic starch-based films are great alternatives for plastics that are based on fossil and petrochemical raw materials. The number of research papers in the field of starch-based materials for packaging purposes is gradually increasing. The starch-based composite or blend has great potential for commercialization in the food packaging field due to it being pollution-free and its simple preparation, desirable functionality, degradability, and low digestibility [4].

Unfortunately, starch materials exhibit some disadvantages that limit their usage as packaging films: moisture sensitivity and low mechanical strength and stability [5]. To improve the properties of thermoplastic starch films, the starch can be modified prior to film preparation by combining it with other biodegradable materials or reinforcing it with active or natural nanoparticles. The enzymatic hydrolysis using α-amylase from *Bacillus amyloliquefaciens* can be used successfully to improve corn starch-based film’s mechanical properties, such as greater elasticity, strength, and stiffness [6]. Adding various natural extracts improves the mechanical properties of the starch-based films, i.e., *Thymus vulgaris* essential oils (TEOs) combined with ethanolic extract of propolis (EEP) increases the elongation at break up to 13.5% [7]. The combination of starch with other polymers, such as polyvinyl alcohol, improves the mechanical and barrier properties of film packaging [8,9,10]. Cellulose nanofibers and nanocrystals can be successfully applied to enhance starch-based materials’ properties [11]. Zeleke et al. have reported the usage of rice straw cellulose fibers to reinforce thermoplastic starch-based films for better industrial applications [12]. According to [12], 10 wt% cellulose nanofibers were the optimum concentration for the composite film, compromising the properties and the transparency. Above 10 wt%, the fiber aggregation in the polymer matrix was observed. Finally, blending plasticized starch with low-density polyethylene allows material development with greater mechanical properties and improved biodegradation for effective environmental pollution control [13].

The packaging materials must fulfill some criteria. The primary function of packaging is to protect the product against external factors. However, the packaging should also communicate accurate information through printed texts and graphics on relevant features of the product, such as ingredients, weight, etc. [14]. Good transparency allows the consumer to see the product packed inside the packaging. Moreover, transparency can be a desired factor for film printing and a legible inscription reading. Therefore, apart from the basic properties of starch films, such as mechanical, barrier, surface, etc., the possibility of printing is crucial. Our previous work showed the potential of overprinting starch films with water-based inks for flexographic technology [15]. The packaging can be overprinted with various techniques, including flexographic, digital, and many more.

Today, ink-jet printing technology is one of the simplest and most popular digital printing techniques, allowing printing on the scale of “short runs” [16]. In the ink-jet printing process, there is a controlled ejection of the liquid on the substrate in a very precise and controlled way [17]. The printing process is contactless, so there is no need for an intermediate carrier to transfer the ink to the substrate. There is no limitation on the substrate type—it can be flexible or rigid—so this technique is suitable for printing on many substrates, including paper, cardboard, textiles, films, glass, metals, etc. In addition, ink-jet printing enables the usage of many different colors and large-format prints. Ink-jet printing has been found to be a widespread technology with broad applications: printing on textiles, production of printing plates, and, finally, functional printing such as 3D printing or printing of so-called printed electronics [18,19,20,21,22]. Ink-jet does not require the usage of expensive printing forms or screens. After a print, the design can be quickly sent to a printing machine and printed on various substrates.

In theory, ink-jet is simple—a print head ejects tiny drops of ink onto the substrate [19]. In practice, ink-jet printing is a complex problem, where implementation of the ink-jet technology needs multidisciplinary skills. The final print quality is influenced by several parameters, mainly the properties of the ink, the substrate, and the printing process parameters. The properties of the ink, such as viscosity and rheology, influence the ink flow in the printhead [16,23]. Next, the surface tension, ink flow, and viscosity influence the drop formation in the printhead [24]. Finally, the properties of the printing substrate affect the final print quality [25,26].

A quick response (QR) code can be used as a data carrier for smart packaging. A QR code is a 2D variant of a barcode, which encodes valuable information. The QR code can be generated simply by using various free tools. A sample QR code (see Figure 1a) was generated using Adobe Express’s QR code creator [27]. The typical QR code is composed of black and white blocks, where they refer to 1 and 0, respectively, representing the bit of information [28]. The white and black squares form a square-shaped grid. Three large squares in the corner allow position detection, size, angle, and shape recognition [29]. This kind of QR code has a poor visual effect. However, by using an appropriate scanning algorithm, a more aesthetic, colorful, and visually pleasant QR code can be generated [30], as shown in Figure 1b–d.

The QR code can be scanned using a scanner or a smartphone with an installed QR code scanner app. The QR code can code useful information about the product in the packaging. QR codes are widely used on a daily basis, and they have been successfully adopted in mobile payments [28], in logistics sorting to allow package tracking and sorting [30], in in situ air analyses [29], and, finally, in food traceability systems to obtain information on the freshness and quality of food products [31,32,33]. Moreover, a QR code can be used to deliver a known amount of a drug [34].

Smart packaging can be developed by combining various printing techniques, materials, and QR codes. An individual QR code label is in a large-scale, rapid, low-cost fabrication process, whereas a unique QR code is overprinted, i.e., one item–one code [35].

This study aims to analyze the print quality of overprinted starch-based films. We have developed two kinds of starch films based on corn and potato starch as a printing base for developing smart, biodegradable packaging to achieve this goal. The properties of starch films were confirmed through mechanical, surface, water vapor transition rate, water solubility, and film color analyses. The developed materials were overprinted with an ink-jet technique with various pictograms and QR codes. Finally, the print quality of the prints was assessed. The starch material with an overprinted QR code giving information about the product forms a unique, smart, and biodegradable packaging for food.

## 2. Materials and Methods

### 2.1. Materials

Corn (NaturAvena, Piaseczno, Poland) and potato (Bio Planet, Leszno, Poland) food-grade starches were used. Glycerol (purity ≥ 99%, CAS 56-81-5) and diiodomethane (purity ≥ 99%, CAS 75-11-6) were purchased from Sigma-Aldrich (Poznań, Poland) and used as received. For the contact angle measurement, water was purified by electrodeionization with the MilliporeSigma Elix Water Purification System (Burlington, MA, USA).

### 2.2. Film Preparation

We previously showed the procedure in our works, for example, see [36]. The reagents (10 g of starch, 5 g of glycerol in 200 g of water) were heated up to 95 °C upon stirring with a mechanical stirrer and cast onto Teflon^®^ plates placed on a K Paint Applicator (RK Print, Royston, UK) equipped with an adjustable micrometer spreader gap set to 3 mm with a constant coating speed (6 m∙min^−1^). The starch films were dried in the climate room for one week in controlled conditions (23 ± 0.5 °C; 50 ± 1% RH).

### 2.3. Film Properties Determination

ATR FT-IR spectra were recorded at room temperature in the 400–4000 cm^−1^ range with a resolution of 4 cm^−1^, using a Nicolet iS5 spectrometer equipped with a Platinum single-reflection diamond ATR module. The FT-IR spectra were analyzed with OMNIC Spectra™ software (series 9.12.968).

The mechanical properties were determined by using a Z010 tensiometer (Zwick-Roell, Ulm, Germany). The measurement was performed according to the ISO 527-1 standard [37]: 15 mm in width, 100 mm in length; the initial distance of the clamps was 50 mm; the stretching speed was 100 mm∙m^─1^. The thickness for the mechanical measurements was performed with a handheld micrometer, with a 0.001 mm resolution and error of ±0.5 mm. The parameters (tensile strength, elongation at break, and Young’s Modulus) were obtained from the stress–strain curves. The measurements were repeated ten times, and the average values were taken as the result.

The surface free energy was calculated using the Owens–Wendt–Rabel–Kaelbe approach [38], based on the contact angle measurement results of water and diiodomethane. The static contact angle was measured with the Drop Shape Analysis System (DSA 30E, Krüss, Hamburg, Germany) in agreement with the ISO 15989 standard [39].

A digital microscope (Keyence VHX-7000, Keyence Corporation, Osaka, Japan) was used to observe the surface structure and the surface roughness determination of developed films. The film’s linear roughness (R_a_ and R_z_) was subsequently evaluated on the Keyence VHX-7000.

The water vapor transmission rate (WVTR) measurements were performed using the MA 210.R (Radwag, Radom, Poland) moisture analyzer. A 54 mm ± 2 mm diameter film sample was placed on an aluminum-sealed probe with 5 g of water and weighed. The WTVR was determined at 45 °C at constant room temperature (23 ± 0.5 °C) and relative humidity (50 ± 1% RH). The measurements were performed manually, and the samples were weighed at 0, 1, and 2 h intervals. The WVTR was calculated using Equation (1):(1)$$WVTR=\frac{∆m}{t\cdot S}$$
where Δ*m* is the difference between the mass of water after 2 and 1 h of measurement, *t* is the time of measurement (2 h), and *S* is the area of the film sample.

The Karl Fisher volumetric titration method (Mettler V30, Beersel, Belgium) was applied for the determination of water content. The sample (approx. 200 mg) was added to a sealed vial. Two component reagents Aquastar^®^-Solvent and Aquastar^®^-Titrant 2 (Supelco, Sigma-Aldrich, Poznań, Poland) were used as a solvent and titrant, respectively.

The solubility of starch-based films was determined according to [40]. Samples of films (approx. 20 mm × 20 mm) were solubilized for 24 h in 50 mL of distilled water (at 25 °C). The solubility was expressed as the percent weight loss of the film strips on soaking, according to Equation (2):(2)$$Solubility\;(\%)=\left(\frac{m_{i}-m_{f}}{m_{i}}\right)\cdot 100\;$$
where *m_i_* is the initial dry mass of the sample and *m_f_* is the final dry mass of the sample after solubilization.

The measurement of WVTR, solubility, and moisture content was repeated in triplicate.

The gloss (in the gloss units, GU) of the prints was measured at 20°, 60°, and 85° geometry conditions with the use of a Micro-Tri-Gloss gloss meter (BYK-Gardner, Geretsried, Germany). Data collection was performed at six different positions of the samples in both directions: cross and machine direction. The reported values are the average of these measurements.

The color of the films was measured using an X-Rite eXact spectrophotometer (Grand Rapids, MI, USA) under the following conditions: D50 luminant, 2° colorimetric observer, and M2 (UVC) filter. The white ink-jet paper was used as the white standard (*L** = 93.73, *a** = 0.79, and *b** = −3.98). The measurement was repeated three times.

### 2.4. Printing

The printing was performed with an eZcolorJet (Graph-Tech, Ft. Pierce, FL, USA) industrial printer equipped with an EPSON I3200 printhead. The printing was performed at 600 dpi with magenta and black UV LED ink (Graph-Tech, Ft. Pierce, FL, USA) printing with 6 pl per 1 drop with max. 42 kHz jetting.

### 2.5. Print Quality Analyses

Optical densities and color values were determined with a SpectroEye spectrophotometer (GretagMacbeth, Zürich, Switzerland). The measurement was performed with the following settings: D50 luminant, 2° colorimetric observer, without polarization filter. The measurement was repeated three times.

The prints were evaluated with a digital portable microscope model Dino-Lite AM 4113T-FVW Premier Handheld Digital Microscope (Dino-Lite Europe, Almare, The Netherlands), to assess print quality, edge sharpness, and microlevel non-uniformity. The selected parts of the prints were captured with 200× magnification and 1280 × 1024 resolution, using DinoCapture 2.0 v 1.5.48 (Dino-Lite Europe, Almare, The Netherlands).

A QR scanning application on a smartphone was used to scan the QR code. In this study, a Mobile Barcode Scanner (v. 5.0.9 (175), Cognex, Natick, MA, USA) and a camera app (Galaxy S21, Samsung, Seoul, Republic of Korea) were used.

All the results are presented as a mean ± SD, where a mean was calculated as an average and SD as a standard deviation using Excel (ver. 2312, Office 365, Microsoft, Redmond, WA, USA).

## 3. Results and Discussion

### 3.1. Properties of Films

Starch films based on potato and corn with 50% glycerol as plasticizers were developed as a printing base. To confirm the quality of starch films, the basic properties were determined, i.e., mechanical properties, surface free energy, water solubility, etc. The measured properties are summarized in Table 1. The mechanical properties, water vapor transition rate, and water solubility determine the use of starch-based materials for packaging purposes. Surface-free energy is the crucial factor influencing the printability of starch-based films. Figure 2 shows the FTIR spectra of both films and the image of the films’ surface.

The ATR-FTIR analyses show no changes in absorption bands, whether corn or potato starch was used for the film development (see Figure 1a). The spectrum shows typical peak characteristics for starch materials: broadband at approx. 3300 cm^−1^, corresponding to O–H stretching vibrations in the hydroxyl group at approx. 2900 cm^−1^, a characteristic peak for CH_2_ stretching vibration, and at approx. 1020 cm^−1^, characteristic of C–O–C stretching vibration [41].

The tensile strength and elongation values at break are comparable to those reported in our previous work [42] and higher than those reported by Dai et al. [43]. The values of mechanical parameters were lower for corn than potato starch films.

The surface roughness, together with the value of surface free energy (SFE) and the polar component of SFE, indicates the possibility of printing. The values of SFE and the polar component of SFE for corn and potato starch films are comparable. At the same time, surface roughness, *R*_a_, is slightly higher for corn starch film.

The barrier properties of films indicate their packaging application. In general, the hydrophilic nature of starch films plasticized with glycerol influences the water vapor permeability [44]. High affinity of glycerol to water promotes the diffusion of water molecules through the films [45]. Potato starch exhibits higher values of WVTR and higher water solubility.

The values of gloss and color properties such as lightness (*L**) and hue parameters (*a**, *b**) are listed in Table 1. For the gloss measurement, geometry with an angle of 85° was used because the gloss of starch materials exhibits lower values than 10 gloss units using a geometry of 60° [46].

The *L** values are similar for corn and potato films. It should be noted that when comparing potato and corn films, both the *a** value and the *b** value for the potato became smaller. Hence, the potato starch is “greener” and “bluer” compared to corn starch film. The color difference between both starch films can be observed with the naked human eye, which is confirmed by the value of the color difference (calculated from the equation: $∆E_{ab\;}^{*}=\sqrt{{\left({∆L}^{*}\right)}^{2}+{\left({∆a}^{*}\right)}^{2}+{\left({∆b}^{*}\right)}^{2}}$, where Δ*L**, Δ*a**, and Δ*b** are the differences between the values of color parameters on corn and potato film), Δ*E* = 1.4.

### 3.2. Printing and Print Quality Assessment

The “print quality” has no strict definition [47], and the assessment depends on the customer’s expectations. The quality of the prints can be assessed by spectrophotometric measurements of color coordinates, the optical density of full-tone coverage, gloss measurement, and adhesion tests [48]. This study determined print quality by determining color parameters *L**, *a**, *b**, and optical densities. The naked human eye control reveals that the prints on the potato starch film are of poorer quality than the overprinted corn or paper samples. The overprinted QR codes and pictograms are shown in Figure 3. The prints were also visually evaluated (Figure 4 and Figure 5).

The print on the paper substrate was used as a reference. Images of the standard QR code and selected pictograms were printed with an ink-jet printer with black and magenta ink. The QR code was printed directly on the surface of the starch film and was scanned using a QR code scanner app installed on a smartphone. The QR code image’s scale, color, and quality critically influence the possibility of scanning. To analyze the deformation of the prints, the digital microscope was used to capture the selected areas on the printouts. To study the difference between images, specific areas on images were selected. Figure 4 shows the deformations of a QR code overprinted on corn starch with magenta and black ink. The overprinted QR code (Figure 3b) was read easily by a smartphone, as shown in Figure 3c. The brighter and less clear QR code block elements (see Figure 4) did not negatively affect its readability. This result was expected because QR codes can still be scanned even with a 30% error [49].

Figure 5 compares the quality of reproduction of printed images and fonts. The letter ‘R’ and the arrow from the compostable logo were analyzed and printed; line width was measured for the image analyses. As it is shown in Figure 4b, the line width depends on the printing substrate, and it is narrower on corn film. The width of the overprinted lines on the starch films is close to the values measured on the paper substrate. The properties and composition of the substrate influence the printed line width [50], which is related to the penetration of the substrate by the ink [51]. Furthermore, it is evident that the ink spread on potato film; therefore, the sharpness and letter quality are worse on potato film.

Table 2 summarizes the *L**, *a**, *b** color parameters and optical density (OD) values. The print color can be assessed by measuring *L**, *a**, *b** (lightness *L**, green–red coordinate *a**, and blue–yellow coordinate *b**). The value of *L** is lower for the overprinted potato and corn starch film than for the overprinted paper. This indicates that the print on starch films is darker than on paper. Moreover, the values of the *a** and *b** parameters are higher.

The optical density of overprinted starch-based films varied from 1.46 to 1.61. The similar values are comparable with the results determined for starch material [15] or other biodegradable polymers, such as PLA [52]. Higher optical density means higher print quality. It is visible that higher density was achieved when printing on the potato film; this may be related to the films’ properties, such as surface free energy and surface roughness. Lower surface roughness and/or higher values of surface free energy of the substrate are related to a higher density due to homogenous ink film formation without the formation of a thicker amount of ink staying in the micro valleys of the substrate surface [53]. Furthermore, the color parameters (*L**, *a**, *b**) exhibited lower values than those for overprinted.

## 4. Conclusions

Printability is one of the significant functional properties of modern materials for packaging purposes. This paper was aimed at the printing performance of starch-based films as a packaging material. To evaluate the printing properties of starch films, corn, and potato starch films with 50% glycerol as a plasticizer were developed. The basic properties of films were measured, i.e., mechanical properties, surface free energy, water solubility, color gloss, etc.

The developed films were used as printing bases and overprinted with ink-jet in technological conditions. The basic print quality of films was assessed with image analyses, color parameters, and optical density. The print quality of overprinted QR codes allows for scanning with a smartphone. Furthermore, the overprinted text and pictograms are legible to the naked human eye. The sharpness and the quality of the lettering are worse on potato film. The line width of selected overprinted elements was wider on potato starch films. At the same time, higher optical densities were measured on potato starch films (1.61 ± 0.06) than on corn starch films (1.46 ± 0.06).

The findings of this study confirm that starch-based films can be used as modern, environmentally friendly packaging and can be overprinted with good quality. The results shown in this paper continue our previous works on the printing and packaging performance of starch-based materials. Our results expand the research on starch-based materials as a source of packaging material. The possibility of good-quality ink-jet printing supports the use of starch-based films for packaging, which makes visible necessary information regarding the packed product.

## Figures and Tables

**Figure 1 materials-17-00455-f001:**
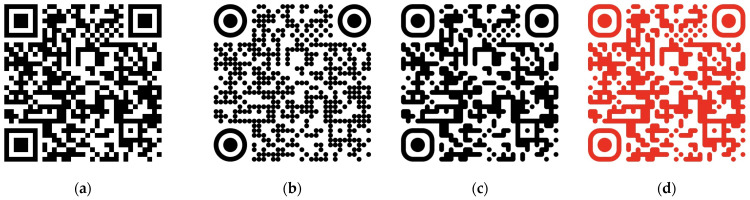
Samples of QR codes, (**a**) basic QR code with conventional square blocks; (**b**–**d**) QR codes with aesthetic blocks.

**Figure 2 materials-17-00455-f002:**
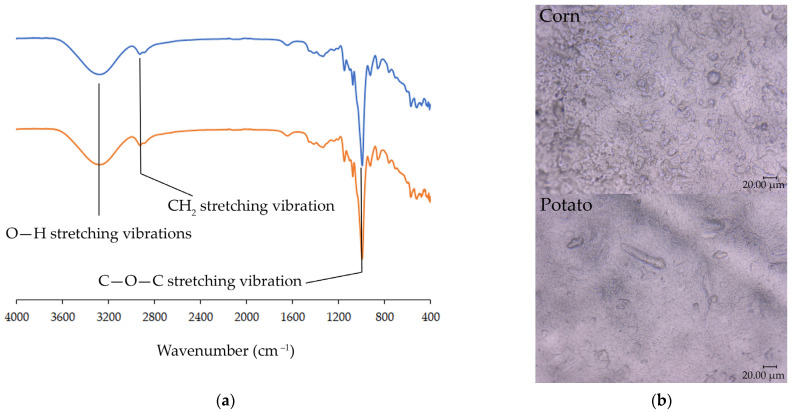
(**a**) FTIR spectrum of corn (blue line) and potato (orange line) film; (**b**) surface images of films at 900× magnification.

**Figure 3 materials-17-00455-f003:**
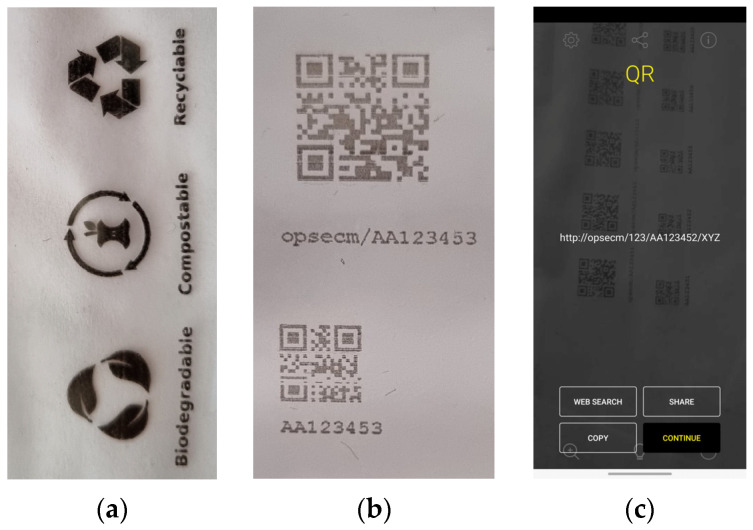
(**a**) Overprinted pictograms on starch-based film; (**b**) overprinted QR-codes; (**c**) screenshot of a smartphone after scanning the QR code.

**Figure 4 materials-17-00455-f004:**
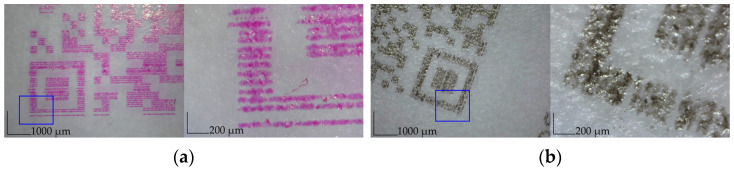
The printed QR codes with magenta and black ink on corn starch: overprinted with magenta ink at magnification 50× and 225× (**a**); overprinted with black ink at magnification 50× and 225× (**b**).

**Figure 5 materials-17-00455-f005:**
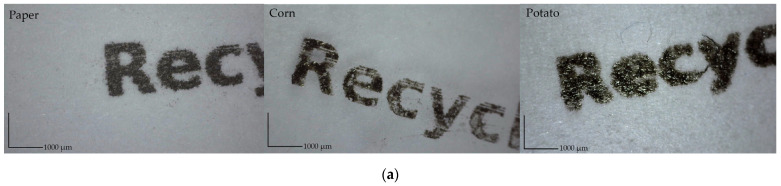
(**a**) Appearance of specific area at 50× magnification; (**b**) appearance of ‘R’ letter in word ‘Recyclable’ at 250× magnification; (**c**) appearance of arrow in word compostable logo at 250× magnification.

**Table 1 materials-17-00455-t001:** Properties of developed films.

Property		Starch Film	
		Corn	Potato
Young’s Modulus (MPa)		42.6 ± 1.2	53.4 ± 1.6
Tensile strength (MPa)		3.47 ± 0.68	4.05 ± 0.60
Elongation at break (%)		33 ± 13	43 ± 6
Thickness (μm)		201 ± 7	80 ± 8
SFE (mJ·m^−2^)		58.67 ± 4.28	59.71 ± 2.39
Polar component of SFE (mJ·m^−2^)		18.89 ± 2.38	17.02 ± 1.15
Surface roughness, *R*_a_ (μm)		1.92 ± 0.16	1.27 ± 0.69
WVTR (%); (mg·cm^−2^·h^−1^)		39; 11.1	43; 12.2
Water content (%)		7.16 ± 0.57	6.48 ± 0.41
Water solubility (%)		18.7 ± 1.4	20.3 ± 1.2
Gloss (°)		4.1 ± 0.9	11.0 ± 3.1
Color	*L**	90.64 ± 0.23	90.72 ± 0.40
	*a**	0.52 ± 0.01	0.84 ± 0.01
	*b**	−2.24 ± 0.08	−3.60 ± 0.04

Values are means ± SD.

**Table 2 materials-17-00455-t002:** Specific ink color components, *L**, *a**, *b**, and optical densities, *OD*, of ink layers printed on corn film.

Sample	*L**	*a**	*b**	*OD*
Paper	40.52 ± 1.31	0.80 ± 0.04	2.08 ± 0.26	0.92 ± 0.02
Corn	21.14 ± 2.78	2.05 ± 0.26	4.29 ± 0.80	1.46 ± 0.06
Potato	16.17 ± 1.81	2.23 ± 0.35	4.08 ± 0.72	1.61 ± 0.09

## Data Availability

The data presented in this study are available upon request from the corresponding author.

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
