# Peer review of "Study on the Printability of Starch-Based Films Using Ink-Jet Printing"

_materials, 2024, doi:10.3390/ma17020455_

Round 1
Reviewer 1 Report
Comments and Suggestions for Authors
This article showed the novel application of starch-based film, as printing substrate were printed with ink-jet printing with quick response (QR) codes, text, and pictograms. It is interesting for expending the research aera for starch based film. The article should be accepted after minor revisions.
1. The performance differences between potato starch film and corn starch film should be analyzed statistically.
2.The strength of the starch film should meet the requirements of the printer, otherwise the printing effect will be poor and it will affect the life of the print head. It is recommended to supplement the basic mechanical properties of the commonly used printing substrates for the printer model, and compare them with the starch film.
3.When the printing volume is large, the increased temperature of the printing equipment caused by the friction of the printing material may affect the strength of the starch film. It is suggested to discuss how to avoid this.
Comments on the Quality of English Language
The text content is excellent in English expression.
Author Response
- The performance differences between potato starch film and corn starch film should be analyzed statistically.
Thank you for your suggestion. In this paper, we have calculated the mean and SD as an average and standard deviation using Excel. The number of repeated measurements was too small to do the statistical analyses properly. The number of repeated measurements was 10, 6, and 3 for mechanical measurements, surface roughness, and other properties, respectively. However, we have added one sentence to clarify (see lines 206-209).
Currently, we don’t have enough samples to perform proper statistical analyses for the paper. But in the future, we will keep this in mind.
2.The strength of the starch film should meet the requirements of the printer, otherwise the printing effect will be poor and it will affect the life of the print head. It is recommended to supplement the basic mechanical properties of the commonly used printing substrates for the printer model, and compare them with the starch film.
3.When the printing volume is large, the increased temperature of the printing equipment caused by the friction of the printing material may affect the strength of the starch film. It is suggested to discuss how to avoid this.
Answer to remarks 2 and 3: We agree with these points. Investigating increased temperature on the mechanical properties of the starch-based film is an interesting research problem. The printing material is exposed to various factors, such as friction, high temperature, mechanical stress, etc. Because of the poorer mechanical properties of starch-based materials compared to conventional plastics. Our research is in basic research and only shows new possibilities for the commercial use of starch films. However, this work aims to analyze the possibility of printing starch materials and the print quality of overprinted starch materials.
In our following paper, we want to optimize the mechanical properties of the films with response surface methodology.
Reviewer 2 Report
Comments and Suggestions for Authors
This article focuses on the research of natural-based materials, namely corn and potato starch, used as base material to create (natural based) plastics for packaging application. The goal is to compare these two in terms of readability of printed patterns (QR codes, text, pictograms) and in terms mechanical properties. Study shows the potential of using these materials as a modern packaging solution.
The article is well written with a sufficient number of experimental techniques, it thoroughly describes the differences between the use of corn and potato starch, it focuses on the properties that are important for practical application. Mainly it focuses on printing application as these materials need to be overprinted by packaging information.
Individual notes and details
LINE 145 – use decimal period or dash. You have both variants on one line (0,001 mm, ± 0.5)
LINE 182 – slightly rephrase the sentence, too much “and”
LINE 200 – better use non-uniformity in microlevel non-uniformity.
LINE 206 – I recommend use the spacers in between numbers and ± sign, for example “42.6 ± 1.2”
Author Response
LINE 145 – use decimal period or dash. You have both variants on one line (0,001 mm, ± 0.5)
Thank You very much for this remark. We have corrected the typos.
LINE 182 – slightly rephrase the sentence, too much “and”
We have divided the sentence into two separate ones.
LINE 200 – better use non-uniformity in microlevel non-uniformity.
Thank You very much for this remark, we have corrected it according to Yours advice.
LINE 206 – I recommend use the spacers in between numbers and ± sign, for example “42.6 ± 1.2”
Thank You for this remark. We have corrected it throughout the paper.